# Fullerenol Quantum Dots-Based Highly Sensitive Fluorescence Aptasensor for Patulin in Apple Juice

**DOI:** 10.3390/toxins14040272

**Published:** 2022-04-12

**Authors:** Hua Pang, Hui Li, Wen Zhang, Jin Mao, Liangxiao Zhang, Zhaowei Zhang, Qi Zhang, Du Wang, Jun Jiang, Peiwu Li

**Affiliations:** 1Oil Crops Research Institute, Chinese Academy of Agricultural Sciences, Wuhan 430062, China; panghua163@163.com (H.P.); zhangwen@oilcrops.cn (W.Z.); maojin106@whu.edu.cn (J.M.); liangxiaozhang@126.com (L.Z.); zhaowei_zhang@126.com (Z.Z.); zhangqi521x@126.com (Q.Z.); wang416929@126.com (D.W.); jiangjun@caas.cn (J.J.); 2Key Laboratory of Biology and Genetic Improvement of Oil Crops, Ministry of Agriculture and Rural Affairs, Wuhan 430062, China; 3National Reference Laboratory for Agricultural Testing (Biotoxin), Wuhan 430062, China; 4Key Laboratory of Detection for Mycotoxins, Ministry of Agriculture and Rural Affairs, Wuhan 430062, China; 5Laboratory of Quality and Safety Risk Assessment for Oilseed Products (Wuhan), Ministry of Agriculture and Rural Affairs, Wuhan 430062, China

**Keywords:** patulin, fullerenol quantum dots, aptasensor, fluorescence

## Abstract

A highly selective and sensitive aptasensor for detecting patulin (PAT) was constructed based on the fluorescence quenching of fullerenol quantum dots (FOQDs) towards carboxytetramethylrhodamine (TAMRA) through PET mechanism. The π-π stacking interaction between PAT aptamer and FOQDs closed the distance between TAMRA and FOQDs and the fluorescence of TAMRA was quenched with maximum quenching efficiency reaching 85%. There was no non-specific fluorescence quenching caused by FOQDs. In the presence of PAT, the PAT aptamer was inclined to bind with PAT and its conformation was changed. Resulting in the weak π-π stacking interaction between PAT aptamer and FOQDs. Therefore, the fluorescence of TAMRA recovered and was linearly correlated to the concentration of PAT in the range of 0.02–1 ng/mL with a detection limit of 0.01 ng/mL. This PAT aptasensor also performed well in apple juice with linear dynamic range from 0.05–1 ng/mL. The homogeneous fluorescence aptasensor shows broad application prospect in the detection of various food pollutants.

## 1. Introduction

Patulin (PAT), also known as coral penicillin, is a secondary metabolite produced by fungi such as *Aspergillus* and *Penicillium*. PAT occurs most often in apples that have been spoiled by mold growth or in products made from spoiled apples, such as apple juice and apple puree [1]. PAT is a neurotoxic mycotoxin, which exerts acute and chronic toxicity to humans and can cause vomiting and nausea. It can change the permeability of the cell membrane and inhibit the synthesis of macromolecules in cells, resulting in the depletion of non-protein sulfhydryl groups and loss of cell activity [2,3]. PAT has toxicological effects on fertility, carcinogenesis, teratogenesis and immunity [4]. Owing to its drastic toxicity, different authorities have established regulatory limits on the level of PAT in food. The maximum limit of PAT defined by the European Union Commission (EU) and China was 50 μg/kg in fruit-based products [5,6], and the maximum daily intake of PAT was setted as 0.4 µg/kg body weight/day by the World Health Organization (WHO) and Food and Agriculture Organization of the United Nations (FAO) [7]. Up to now, various kinds of protocols have been developed for PAT detection, including thin layer chromatography (TLC), High-performance liquid chromatography (HPLC), high-performance liquid chromatography tandem mass spectrometry (HPLC-MS), micellar electrokinetic chromatography (MEKC) and so on [8,9,10]. Although these methods exhibit high reliability and accuracy, some of them are time-consuming and require expensive instruments and highly skilled operators, which makes them unsuitable for in situ analysis. Therefore, it is urgent to establish simple and fast analytical methods for PAT detection in fruit products to ensure human health and consumption safety.

Aptamers are short synthetic single-stranded oligonucleotides or peptide synthesized by systematic evolution of ligands by exponential enrichment (SELEX) method, which is mainly repeated in the binding, separation, amplification and purification steps [11,12,13]. In the past few years, with the unique advantages of cost-effective and ease of synthesis, high affinity, good stability and simple modification, aptamers have attracted increasing attention and been widely used in biosensing applications for heavy metal ions, antibiotics, drugs, protein and even whole cells detection [14,15,16,17]. Recently, Wu et al. screened and identified a 40-mer aptamer for PAT and developed a colorimetic aptasensor for PAT detection based on an enzyme-chromogenic substrate system in 2016 [18]. Afterwards, electrochemical analysis method based on PAT aptamer were also constructed for detecting PAT [19]. Although some progress has been made in establishing PAT aptasensors, it is still very important to set up new biosensors for PAT with better performance.

Due to simplicity and accessibility of fluorescent labels with diverse spectral characteristics for optical signal transduction, fluorescence-based assay methods are readily becoming the most widely used technique in the field of sensing [20]. Various fluorescence-based biosensing platforms have been developed for different applications such as food safety, environmental monitoring, drug delivery and bioimaging etc [21,22,23,24]. With the rapid development of nanotechnology in material science, some new fluorescent labeling materials and nanomaterial-based quenchers with excellent optical character have emerged and attracted a lot of attention [25]. It has been demonstrated that fullerenols quantum dots (FOQDs) with good fluorescence quenching performance are very suitable for application in the field of biosensing [26]. It has also been verified that π-π stacking interaction existed between FOQDs and biological molecules [27,28].

In order to achieve highly sensitive fluorescence detection of PAT, the excellent fluorescence-quenching ability of FOQDs and the high affinity of PAT aptamer towards PAT were combined. When TAMRA-labeled PAT aptamer and FOQDs were mixed, the π-π stacking interaction between PAT aptamer and FOQDs closed the distance between energy donor TAMRA and energy acceptor FOQDs. The fluorescence of TAMRA was quenched. However, in the presence of PAT, the specific binding of PAT aptamer and PAT resulted in the conformational change of aptamer. Thereby the π-π stacking interaction between PAT aptamer and FOQDs was plainly reduced and the distance between TAMRA and FOQDs was widened. In this case, the fluorescence recovery of TAMRA was observed in a PAT concentration-dependent manner. The proposed PAT fluorescence aptasensor also worked well in apple juice samples. This highly sensitive and selective fluorescence aptasensor exhibit wider application prospect in detecting other mycotoxins and so on.

## 2. Results and Discussions

### 2.1. Fluorescence Aptasensor Development for PAT

The PAT aptasensor was developed on the basis of fluorescence quenching and recovery between TAMRA and FOQDs, as shown in Figure 1. It has been reported that strong π-π stacking interaction existed between ssDNA aptamer and FOQDs. However, after the specific binding of ssDNA aptamer with its target, the π-π stacking interaction was greatly weakened because of the conformational change of PAT aptamer. In our design, the π-π stacking interaction closed the distance between TAMRA and FOQDs, resulting in the fluorescence quenching of TAMRA through PET mechanism. After PAT was introduced into the TAMRA-PAT aptamer-FOQDs PET system, PAT aptamer was inclined to bind with PAT and its conformation changed, giving rise to weak π-π stacking interaction between changed conformation of PAT aptamer and FOQDs. Thus, TAMRA was far away from FOQDs and PET process inhibited. The fluorescence of TAMRA recovered and the fluorescence recovery was positively correlated with the concentration of PAT.

### 2.2. Properties Characterization of the FOQDs

FOQDs was purchased and used to realize the above design. It has a particle size of 2 nm. Fourier Transform Infrared (FT-IR) spectrum of FOQDs was shown in Figure 2b, indicating the typical absorptions of fullerenols. It can be seen an intense broad O-H band around 3490 cm^−1^, and three characteristic bands around 1620, 1410 and 1090 cm^−1^ corresponding to C=C, C-O-H and C-O absorption, respectively. Energy Dispersive Spectroscopy (EDS) results (Figure 2c) which indicated that C elements and O elements were both on the surface of FOQDs, further confirmed that FOQDs had abundant hydroxyl groups [29]. Na elements might be attributed to the use of NaOH in the process of FOQDs synthesis from fullerene [30]. It was clearly demonstrated that there were a large amount of hydroxyl groups to form a hydrophilic surface on the surface of FOQDs, which greatly reduced the particle-to-particle interactions and improved its dispersion in water [31,32].

### 2.3. Construction of the PAT Aptasensor

Firstly, the concentration of TARMA-labeled PAT aptamer was optimized. It was indicated in Figure 3a that the fluorescence of TAMRA-labeled PAT aptamer enhanced with the concentration increased from 10 nM to 60 nM. At the concentration of 60 nM, the fluorescence intensity was enough to establish the fluorescence aptasensor for PAT detection. Although higher concentration could result in the increasing of the fluorescence intensity, the high concentration would also reduce the detection sensitivity. So, 60 nM of TAMRA-labeled PAT aptamer was used in this fluorescence aptasensor. In order to study the energy transfer process between TAMRA and FOQDs, different concentrations of FOQDs were mixed with TAMRA-labeled PAT aptamer at a fixed concentration of 60 nM. They were incubated in HEPES buffer (50 mM, 5 mM MgCl_2_, 120 mM NaCl, pH 7.4). It could be seen from Figure 3b that the fluorescence of TAMRA-labeled PAT aptamer was quenched with 85% maximum quenching efficiency, depending on the concentration of FOQDs. To rule out non-specific fluorescence quenching between FOQDs and TAMRA, the fluorescence of TAMRA at a final concentration of 60 nM mixed with 70 μg/mL of FOQDs in HEPES buffer was measured. From Figure 3c, it indicated that there was nearly no non-specific fluorescence quenching. Therefore, the fluorescence quenching of TAMRA was attributed to the strong π-π stacking interaction between FOQDs and PAT aptamer, which closed the distance between TAMRA and FOQDs resulting in PET [27]. So 70 μg/mL of FOQDs was chosen for the following experiments. Then TAMRA-labeled PAT aptamer with a concentration of 60 nM was incubated with FOQDs in a concentration of 70 μg/mL, and the fluorescence intensities were measured after different incubation time from 0 to 30 min to obtain the time-dependent fluorescence intensities. As indicated in Figure 3d, 5 min was enough to achieve the quenching equilibrium. A 30 min incubation period was used for the fluorescence quenching process in the next fluorescence recovery experiment to ensure that a quenching equilibrium could be achieved.

### 2.4. PAT Detection in Aqueous Buffer Solution

As illustrated in Figure 1, in the presence of PAT, the PAT aptamer was inclined to bind with PAT and its conformational was changed. There was very weak π-π stacking interaction between conformation-changed PAT aptamer and FOQDs, as TAMRA was far away from FOQDs, resulting in the inhibition of PET process. The fluorescence of TAMRA recovered and had a positive correlation with PAT concentration, as shown in Figure 4a. It was indicated that the fluorescence recovery of TAMRA linearly corresponded to the concentration of PAT in the range of 0.02 ng/mL^–1^ ng/mL, with the detection limit of 0.01 ng/mL (Figure 4b). Compared to other reported PAT biosensors [18,33,34,35], the present sensor performed better as indicated in Table 1, showing great potential for lower concentration of PAT detection in agricultural products. This improvement depended upon the good fluorescence quenching ability of FOQDs, with negligible non-specific fluorescence quenching to non-labeled TAMRA. Other mycotoxins, including zearalenone (ZEN), fumonision B1 (FB_1_), deoxynivalenol (DON), T-2 toxin and aflatoxin B1 (AFB1) were introduced into the TAMRA-PAT aptamer-FOQDs PET mixture to investigate the specificity of the aptasensor for PAT. In Figure 5, it could be found that all the other mycotoxins cause negligible fluorescence alteration of TAMRA except PAT, which fully confirmed the superior specificity of the aptasensor towards PAT resulting from the high binding affinity of PAT aptamer.

### 2.5. PAT Detection in Apple Juice

It is absolutely essential to monitor the concentration of PAT in apple juice to ensure the safety of consumption and the health of consumers. PAT detection was also realized in apple juice that is 100-fold diluted with HEPES buffer. Figure 6a indicates that the fluorescence of TAMRA was linearly correlated to PAT concentration ranging from 0.05 ng/mL to 1 ng/mL, with a detection limit of 0.03 ng/mL (Figure 6b). Subsequently, a standard addition experiment was carried out to examine the practical application of this PAT aptasensor in apple juice. Satisfactory recoveries were obtained from 95% to 106% in Table 2. It firmly demonstrated that this aptasensor constructed with FOQDs as energy acceptor had great potential application.

## 3. Conclusions

In conclusion, a highly sensitive and selective PET aptasensor for PAT detection has been developed. FOQDs exhibited superior fluorescence-quenching ability with 85% maximum quenching efficiency. What’s more, the non-specific fluorescence quenching was almost negligible. Combined with the high specificity aptamer of PAT, this aptasensor performed well both in aqueous buffer solution and apple juice samples. This homogeneous aptasensor also had the advantages of simplicity and high efficiency, indicating broad application prospects in detecting various food pollutants, such as other mycotoxins.

## 4. Materials and Methods

### 4.1. Materials

TAMRA-labeled aptamer sequences for PAT used in this study were purchased from Sangon Biotech (Shanghai, China) Co., Ltd. Wuhan Synthesis Department. TAMRA-labeled aptamer sequences are as follows: 5′-ggC CCg CCA ACC CgC ATC ATC TAC ACT gAT ATT TTA CCT T-TAMRA-3′.

Patulin, potassium phosphate monobasic (KH_2_PO_4_), sodium phosphate dibasic (Na_2_HPO_4_), magnesium chloride (MgCl_2_), potassium chloride (KCl), HEPES sodium salt, FOQDs and other mycotoxins: ZEN, FB1, DON, T-2, AFB1 were purchased from Sigma-Aldrich (St. Louis, MO, USA). Milli-Q water obtained from a Milli-Q system (Millipore, Bedford, MA, USA) was used in the experiment. Apple juice (purity: 100%) was bought in a local supermarket.

### 4.2. Instrumentation

The size and morphology were recorded by a HITACHI H-7000FA transmission electron microscope (Tokyo, Japan). FT-IR spectrum of FOQDs was measured on a Thermo Fisher Scientific Nicolet iS10 FT-IR Spectrometer (Waltham, MA, USA). Fluorescence spectra were recorded using a Hitachi F4500 fluorescent spectrophotometer (Tokyo, Japan).

### 4.3. Quenching Measurements

Different concentrations of FOQDs were used to incubate with TAMRA-labeled PAT aptamer at a fixed concentration of 60 nM. The concentration of FOQDs were set at 0, 10 μg/mL, 20 μg/mL, 30 μg/mL, 40 μg/mL, 50 μg/mL, 60 μg/mL, 70 μg/mL, 80 μg/mL, 90 μg/mL, 100 μg/mL, 150 μg/mL, 200 μg/mL. They were incubated in HEPES buffer for 1 h and the fluorescence signals were obtained by excitation at 550 nm. Then TAMRA-labeled PAT aptamer with a concentration of 60 nM was incubated with FOQDs in a concentration of 70 μg/mL. The time-dependent fluorescence signals were measured after different incubation time from 0 to 30 min.

### 4.4. PAT Detection in Buffer Solution

Various concentrations of PAT (0, 0.01 ng/mL, 0.02 ng/mL, 0.03 ng/mL, 0.06 ng/mL, 0.09 ng/mL, 0.1 ng/mL, 0.2 ng/mL, 0.4 ng/mL, 0.6 ng/mL, 0.8 ng/mL, 1.0 ng/mL) were respectively mixed with TAMRA-labeled PAT aptamer (60 nM) for 1 h at room temperature. Then, FOQDs were added individually at a concentration of 70 μg/mL and incubated for another 30 min. The fluorescence signals were recorded. To investigate the selectivity of this PET aptasensor, other toxins including ZEN, FB1, DON, T-2 and AFB1 were introduced into the TAMRA-PAT aptamer-FOQDs PET mixture. ZEN, FB1, DON, T-2 and AFB1 with a concentration of 0.4 ng/mL were incubated with TAMRA-labeled PAT aptamer with a concentration of 60 nM in HEPES buffer at room temperature for 1 h, respectively. A total of 70 μg/mL FOQDs were added individually and incubated for 30 min. Then, the fluorescence intensities were measured.

### 4.5. PAT Detection in Apple Juice Samples

The apple juice samples were purchased from the local market of Wuhan, China. In order to detect PAT in the apple juice, the apple juice was diluted 100 times with HEPES buffer to avoid matrix interference. The assay procedure was the same as that in the HEPES buffer. Standard addition method was carried out by adding different PAT (0.40 ng/mL, 0.80 ng/mL, 1.00 ng/mL) into PAT-free apple juice samples and then detected the concentration of PAT by this fluorescence aptasensor.

## Figures and Tables

**Figure 1 toxins-14-00272-f001:**
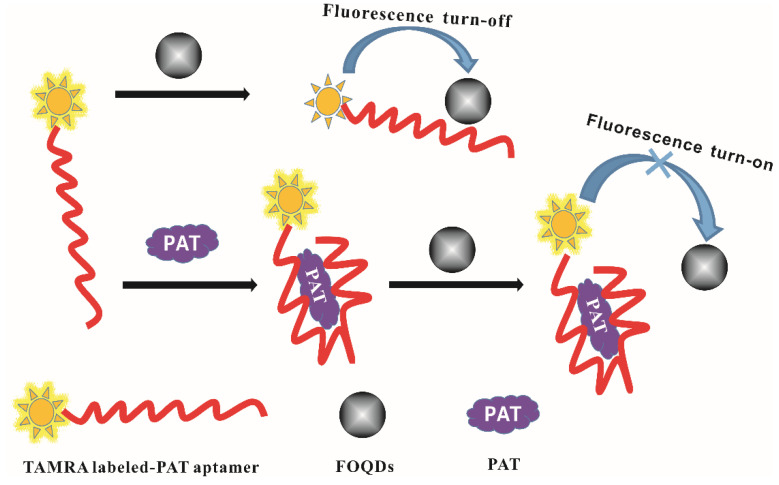
Schematic diagram of the fluorescence aptasensor for PAT detection based on fluorescence turn-off and turn-on from TAMRA to FOQDs. When TAMRA labeled-PAT aptamer was incubated with FOQDs, fluorescence turn-off was observed by the luminescence quenching caused by FOQDs towards TAMRA through PET mechanism. While in the presence of PAT, the aptamer was inclined to bind with PAT and its conformation changed. TAMRA was far away from FOQDs and PET process was inhibited, resulting in the fluorescence turn-on of TAMRA.

**Figure 2 toxins-14-00272-f002:**
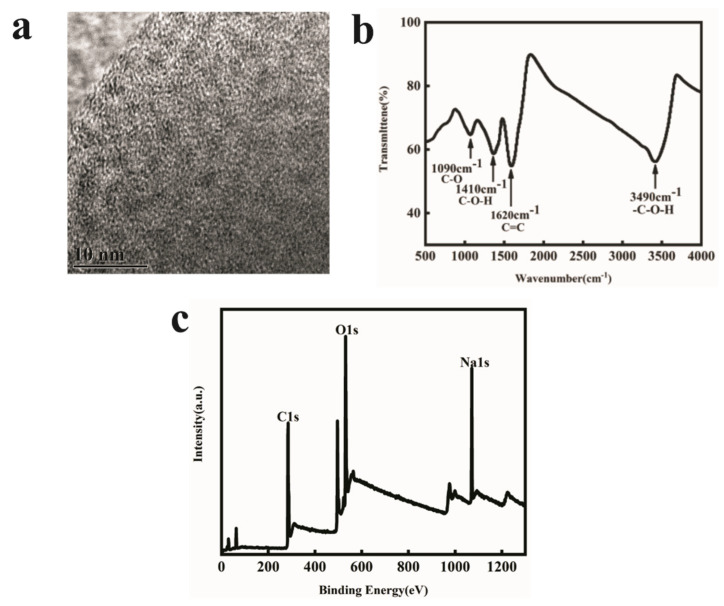
(**a**) TEM spectrum, (**b**) FT-IR spectrum and (**c**) EDS spectrum of FOQDs.

**Figure 3 toxins-14-00272-f003:**
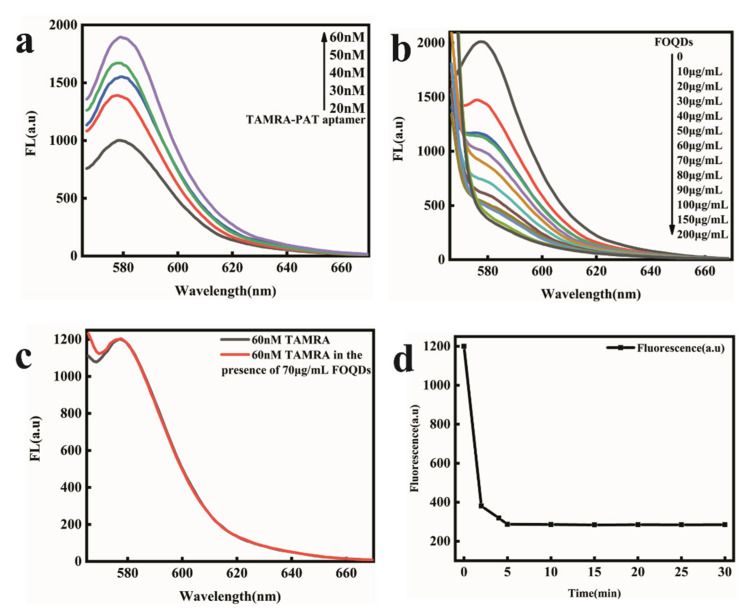
(**a**) Fluorescence spectra of TAMRA labeled-PAT aptamer at different concentrations ranging from 20 nM to 60 nM. (**b**) Luminescence quenching of TAMRA-PAT aptamer (60 nM) withdifferent concentrations of FOQDs (0, 10 μg/mL, 20 μg/mL, 30 μg/mL, 40 μg/mL, 50 μg/mL, 60 μg/mL, 70 μg/mL, 80 μg/mL, 90 μg/mL, 100 μg/mL, 150 μg/mL, 200 μg/mL). (**c**) Fluorescence spectra of TAMRA (60 nM). (**d**) Time dependence of the luminescence quenching efficiency of 60 nM TAMRA-labeled PAT aptamer and 70 μg/mL FOQDs.

**Figure 4 toxins-14-00272-f004:**
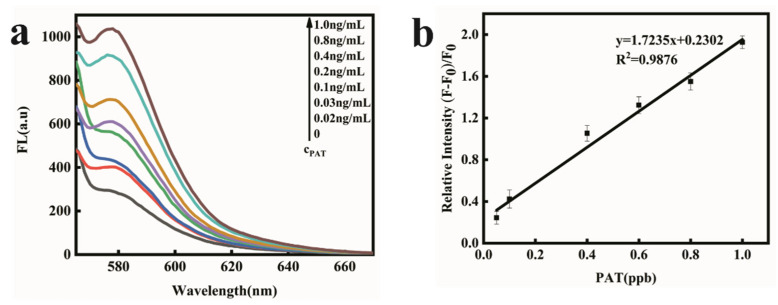
(**a**) Fluorescence recovery spectra in the presence of increasing concentration of PAT (0, 0.02 ng/mL, 0.03 ng/mL, 0.1 ng/mL, 0.2 ng/mL, 0.4 ng/mL, 0.8 ng/mL, 1.0 ng/mL). (**b**) The fluorescence recovery (at 580 nm) linearly corresponded to the concentration of PAT in the range from 0.02 to 1 ng/mL in HEPES buffer.

**Figure 5 toxins-14-00272-f005:**
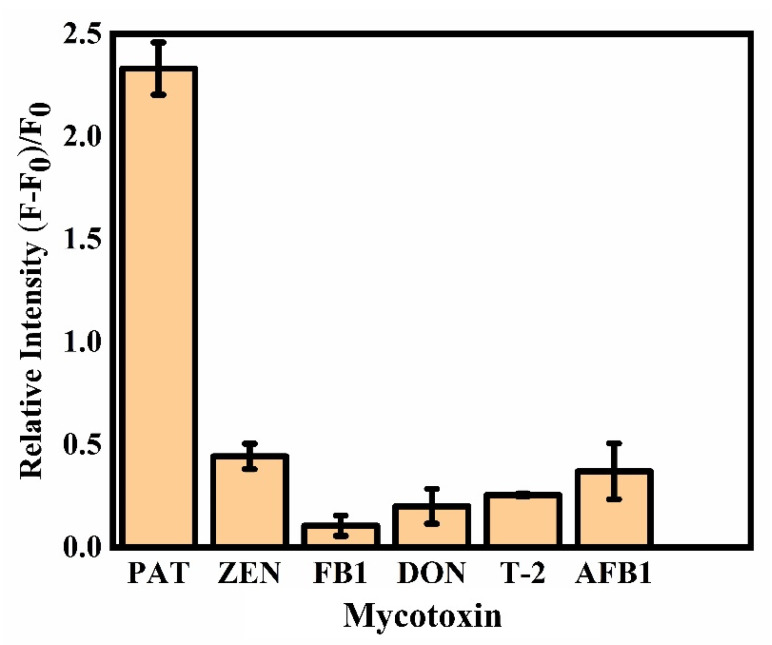
Selectivity experiment of developed aptasensor against other mycotoxins at a concentration of 0.4 ng mL^−1^.

**Figure 6 toxins-14-00272-f006:**
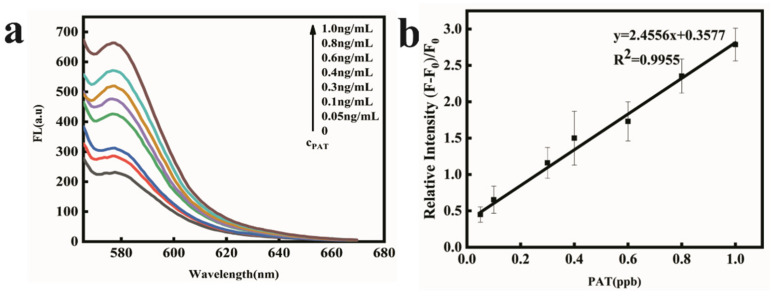
(**a**) Fluorescence recovery spectra with increasing concentration of PAT (0, 0.05 ng/mL, 0.1 ng/mL, 0.3 ng/mL, 0.4 ng/mL, 0.6 ng/mL, 0.8 ng/mL, 1.0 ng/mL). (**b**) Fluorescence recovery linearly corresponded to PAT concentration in the range of 0.05 to 1 ng/mL in 100-fold diluted apple juice with HEPES buffer.

**Table 1 toxins-14-00272-t001:** Comparison with other biosensors for PAT detection.

Method	LINEAR RANGE	Limit of Detection	Detection Time	References
Fluorometric aptasensor based on magnetized graphene oxide and DNase I-assisted target recycling amplification	0.5–30 ng/mL	0.28 ng/mL	~2 h	[33]
Quartz crystal microbalance sensor based on molecularly imprinted sol-gel polymer	7.5 × 10^−3^ μg mL^−1^–6 × 10^−2^ μg mL^−1^	3.1 × 10^−3^ μg mL^−1^	~4 h	[34]
colorimetric method based on aptamer and gold nanoparticles	50–2500 pg mL^−1^	48 pg mL^−1^	~2 h	[18]
Phosphorescent nanosensor based on surface molecularly imprinted polymer capped Mn-doped ZnS quantum dots	0.43–6.50 μmol L^−1^	0.32 μmol L^−1^	2 h	[35]
Fluorescence aptasensor based on FOQDs	0.02–1 ng/mL	0.01 ng/mL	1.5 h	This work

**Table 2 toxins-14-00272-t002:** Analytical and recovery performance of developed aptasensor for PAT detection.

Apple Juice Sample	PAT Added (ng/mL)	PAT Founded (ng/mL)	Recovery(%)	RSD(%) *n* = 3
1	0.40	0.38	95.0	5.3
2	0.80	0.85	106.3	3.1
3	1.00	1.03	103.0	5.0

## Data Availability

Not applicable.

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
