# Peer review of "Fullerenol Quantum Dots-Based Highly Sensitive Fluorescence Aptasensor for Patulin in Apple Juice"

_toxins, 2022, doi:10.3390/toxins14040272_

Round 1
Reviewer 1 Report
Kindly change the title of the manuscript for better clarity for readers.
Please include references at appropriate places in the main text. Several places in the manuscript need references.
Eg: limit of PAT defined by the European Union Commission (EU) and China was 50 μg/kg in fruit-based products (Reference).
And the maximum daily intake of PAT was setted as 0.4 μg/kg body weight/day by the World Health Organization (WHO) and Food and Agriculture Organization of the United Nations (FAO) (Reference).
of cell activity [2,3]. Here reference no 3 is not appropriate. Please replace reference 3 with “The effects of mycotoxin patulin on cells and cellular components. Trends in Food Science & Technology, 83, 99–113”. This review paper covers the cellular damages caused by PAT.
TAMRA-labeled PAT aptamer, that is, 60 nM. What is the reason for the use of 60 nM of aptamer?
Similarly, how do authors determine the 70 μg/mL of FOQDs for analysis?
The time-dependent fluorescence intensities were obtained by incubating a fixed concentration of TAMRA-labeled PAT aptamer (60 nM) with FOQDs in a concentration of 70 μg/mL. How?
Does the abbreviation need a full name at first occurrence? Eg: ZEN、FB1、DON,、T-2 and AFB1
As per the regulatory organization (EU and others) maximum limit of PAT in foods is 50 µg/kg, then what is necessary to detect the PAT concentration at 0.02 ng/mL? What is the significance to detect at a lower concentration than the maximum limit?
Entire manuscripts have to be checked for grammar and typo errors.
Eg: 0、0.01 ng/mL、0.02 ng/mL、.0.03 ng/mL、0.06 ng/mL、0.09 ng/mL、
fruit-based products. And the maximum daily
tamer (60 nM) in HEPES buffer, respectively, and
Figures are not clear kindly improve the quality (except fig.4).
Aptamer sequences are as follow: 5’-ggC CCA ACC CgC ATC ATC TAC ACT gAT ATT TTA CCT T-3’. On what basis this aptamer sequence was chosen for this study?
How did the authors determine the affinity, specificity, and selectivity of aptamer with PAT? Kindly include it in the main text.
Kindly include a comparative table with previously published data in the related field with present findings. This table will show the significance of the present study novelty, by detection time, limit, and accuracy.
Reviewer 2 Report
A very insightful piece of work. Well written and well communicated. However, there are some few suggestions to help improve the quality of the manuscript. Find detailed comments attached.
A very interesting and relevant subject area which is definitely innovative. I wish to suggest the following points to further strengthen this manuscript
- The term TAMRA needs to be expanded
- Line 51…’ Wul et al “….please add the year of publication
- Line 57..”Due to simplicity and accessibility
- The objective of this study must be clearly stated in the introduction
- Scheme 1 is NOT so legible and clear…..kindly work on it to improve the clarity
- Line 169….data were presented as average and RSD
- Authors should high light more on the uniqueness of this research in the introduction
- 4…Please label the x-axis
- Can this technique be used to detect the presence of other mycotoxins??
- Results obtained can be compared with the accuracy of other existing techniques to make it more interesting
Reviewer 3 Report
In the manuscript, the authors developed a fluorescence aptamer biosensor for patulin (PAT).
The authors have declared that their FOQDs-based aptasensor highly sensitively detected PAT. The authors consider their findings represent an advance in knowledge of PAT detection. However, we raised some questions that have to be addressed in this manuscript.
Below are some notes.
The title reflects the contents of the manuscript but is too long. We suggest to revising the original title.
The figures require improvements before their publication. All the figures should be of high quality to correct reading/understanding.
All the M & M should be described with sufficient detail to allow others to replicate on your results.
The discussion of the results is a bit low detailed. We recommend that you improve the discussion for a more accessible one.
Lines 46-47: “Aptamers are short synthetic single-stranded oligonucleotides or peptide synthe-
sized by systematic evolution of ligands by exponential enrichment (SELEX) method.”
The authors should add more information on SELEX. Please provide sufficient background and include references.
Lines 59-61: “Various fluorescence-based biosensing platforms have been developed for different applications such as food safety, environmental monitoring, drug discovery and bioimaging etc. “
Do you have any citations backing this statement up? If so, please include them.
Lines 64-67: “It has been demonstrated that fullerenols quantum dots (FOQDs) have several advantages over conventional organic quenchers such as high quenching efficiency, making them particularly suitable for biosensing applications.”
The authors should add more information on FOQDs. No reference to any source. Please, complete the information, according to some literature.
Lines 97-98: “Scheme 1. Schematic illustration of the aptasensor for PAT detection based on aptamer-bridged fluorescence turn-off and turn-on from TAMRA to FOQDs.”
“Scheme 1” should be “Figure 1”. Moreover, there is a lack of detail in that statement. It should be re-written. Please, reformulate it.
Lines 99-108:
The section “2.2. Properties Characterization of the FOQDs” in the manuscript is not presented clearly. For example, the authors state that “The FT-IR spectra of….” but do not explain what does this mean. Please add more information on FT-IR spectra.
Moreover, what do you mean with "XPS"? Please add more information on it. It is not clear and should be re-written.
We recommend that you improve this section for a more accessible one.
Lines 117-123; 143; 150: “the HEPES buffer (50 mM, 5 mM MgCl2, 120 mM NaCl, pH 7.4)” is already present in the paper; it’s a repeat … please modify it to avoid redundancies.
Line 147: “TAMTA” should be “TAMRA“.
Lines 158- 159:
“Other interfering toxins, including ZEN、FB1、DON、T-2 and AFB1 were added individually into the TAMRA-PAT aptamer-FOQDs”.
What do you mean with "ZEN、FB1、DON、T-2 and AFB1"? Please add more information on it. It is not clear and should be re-written.
Lines 224-226, 4.2. Instrumentation:
These sentences don’t sound good. Please modify them to avoid redundancies.
Lines 250-251:
The authors state that “the apple juice was 100-fold diluted with HEPES binding buffer” but do not explain what does this mean. Please explain to readers why “the apple juice was 100-fold diluted with HEPES binding buffer”. Please add more information on it.
Figure 5, caption: The authors state that “Experiments with the addition of increasing concentration of PAT were conducted in HEPES buffer”, but do not explain what does this mean. Did you refer to “the apple juice” diluted with HEPES binding buffer? The sentence requires improvements before its publication.
I hope this helps.
Round 2
Reviewer 3 Report
The figures still require improvements before their publication. Please, improve the quality of your figures.
The manuscript should be checked by a native speaking English proof-reader.
